# Validity and reliability of sprint force-velocity profiling in elite football: Comparison of MySprint, GPS, and radar devices

**Robert A. Stockdale**[1]*, **Thomas Dos'Santos**[2], **Kevin McDaid**[3], **Philip Nagy**[1], **Christopher J. Gaffney**[1], **Timothy J. Barry**[1]

1 Lancaster Medical School, Health Innovation One, Sir John Fisher Drive, Lancaster University, Lancaster, United Kingdom, 2 Department of Sport and Exercise Sciences, Manchester Institute of Sport, Manchester Metropolitan University, Manchester, United Kingdom, 3 Department of Computing and Mathematics, Institute of Technology, Dundalk, Ireland

* r.stockdale@lancaster.ac.uk

## Abstract

The aims were to examine the validity and within-session inter-trial, intra- and interrater reliability of sprint force-velocity profiling (FVP) techniques in elite football. Twelve elite youth football players from an English Premier League academy participated in this study. A 30-m maximal effort linear sprint testing protocol was conducted, simultaneously measured using the MySprint app, GPS units, and radar device to derive theoretical maximal horizontal force ($F_0$), theoretical maximal running velocity ($V_0$), and the overall orientation of the profile ($FV_{slope}$). There were significant differences in $F_0$, $V_0$, and $FV_{slope}$ ($p < 0.05$) between the MySprint app and radar device, with large effect sizes ($\varepsilon^2 = 0.683$, 0.513, and 0.482), but not in $F_0$ or $V_0$ ($p > 0.05$), between the GPS and radar devices. There were no significant differences in $F_0$ or $V_0$ ($p > 0.05$) between each players' sprint repeats for all equipment types, with these effect sizes ($r_{rb}$): MySprint (0.047, 0.523), GPS (0.236, −0.163), radar (0.785, −0.777). RMSE and CV values for $F_0$ and $V_0$ were 0.72–1.20 N/Kg and 3.76–9.59%, 0.21–0.40 m/s and 1.48–2.64%, respectively, with improved reliability and agreement in $V_0$ vs. $F_0$. There was no significant difference in $F_0$ or $V_0$ ($p > 0.05$), with medium-to-large effect sizes ($r_{rb} = −0.242$ and −0.636) when the MySprint analysis was repeated by the same rater. RMSE and CV values for $F_0$ and $V_0$ were 0.19 N/Kg and 2.9%, 0.07 m/s and 0.54%. There was no significant difference in $F_0$ or $V_0$ ($p > 0.05$) between the two raters' MySprint analysis, and ICC demonstrated excellent agreement ($F_0 = 0.986$, CI = 0.950–0.996, and $V_0 = 0.988$, CI = 0.957–0.997). This study reports high intra- and inter-rater reliability when using the MySprint app to derive FVP's from 30-m maximal sprint testing whilst indicating GPS as the most valid and reliable force-velocity profiling technique against the reference radar device.

**Data availability statement:** All relevant data are within the article and its supporting information files.

**Funding:** RAS (lead/corresponding author) was employed part-time by Burnley F.C. as a First Team Research & Development Assistant during the period of data collection. This role was salaried, but the employment concluded in June 2023, prior to the preparation of this manuscript. No specific financial or material support was provided for this study. The funders had no role in study design, data interpretation, manuscript preparation, or the decision to publish. A member of Burnley F.C. staff (the lead academy sports scientist) contributed to data collection and analysis using the MySprint app. The research was conducted independently, and to confirm, the authors received no specific funding for this work.

**Competing interests:** The authors have declared that no competing interests exist.

## Introduction

Sprinting is a crucial element of football performance [1], contributing to positive outcomes including goal scoring and defensive actions [2], and displaying increased frequency in the English Premier League [3]. FVP outputs are vital for evaluating acceleration performance [4] and assessing potential hamstring injury risk [5,6]. Hamstring muscle injuries are the most common injuries in elite football, affecting 22% of players per season, with recurrence rates of between 14% and 63% in the same playing season following initial injury, leading to prolonged absences [7,8]. The sprint force-velocity profile (FVP), describing the linear relationship between horizontal force and velocity [9], can assess sprint performance in elite football [10–12]. Elite male footballers' maximal sprint velocity typically peaks between 20–40-m, indicating that FVP protocols should be conducted ~30 meters to capture maximal performance and ensure valid and reliable FVP outputs [13,14]. Studies have associated reductions in force from this profiling with retrospective [15–17] and potential prospective hamstring muscle injury risk [6,18,19], emphasising its relevance for both performance and injury prevention.

FVP has previously been confined to laboratories, using force platforms or instrumented treadmills to directly measure horizontal and vertical ground reaction forces, alongside horizontal velocity, throughout the sprint [20]. Whilst accurate, these methods are costly, time-intensive, and, in the case of instrumented treadmills, can disrupt natural sprinting due to balance issues caused by incorrect setting of motor torque [21]. Field-based methods measuring time to displace a specific distance (timing photocells) or instantaneous velocity (radar/laser systems), were previously employed for FVP [22]. Recently, the Simple Method of FVP [9] has arisen, leveraging an inverse dynamics approach to indirectly assess the kinematics and kinetics of the athlete's centre of mass during linear sprints [23]. This method has been reported as an accurate, reliable, and valid approach for determining sprint FVP's in field conditions, overcoming the traditional methods' limitations [24]. Smartphone applications (MySprint) and global positioning systems (GPS) integrating this method, are being used for FVP due to enhanced accessibility, especially in practical settings, although it is important to also interrogate these methods' validity against reference devices [25,26].

Ensuring high intra- and inter-rater reliability is essential in FVP, particularly through 2D camera-based analysis (MySprint) which requires manual frame identification, as measurement errors can significantly impact integrative indexes such as theoretical maximal horizontal force ($F_0$), theoretical maximal running velocity ($V_0$), and the overall orientation of the profile ($FV_{slope}$) [27], which underpin the linear relationship between horizontal force and velocity. Studies highlight the need for intra-rater reliability, standardisation, and familiarisation to address sources of error and maintain methodological rigor, with additional factors including age, sex, and playing level influencing FVP outcomes [18]. Standardising protocols, including sprint start positions [15,16], enhances the reliability and validity of FVP outputs, particularly as inaccuracies in split times (calculated by manual frame identification) can amplify integrative index error. Rigorous examination and

reporting of error sources differentiates between the reliability of the profiling concept, methodology, and input data [27]. Intra-rater reliability identifies potential errors and consistency in scores within the same rater, whilst inter-rater reliability assesses whether different practitioners can consistently determine profiles, especially important in elite football.

The MySprint app has been concurrently validated for FVP, using radar devices as the reference method [28]. The original app (version 1.7), designed for 40-m sprints, showed strong agreement with radar-based measurements, using Pearson's correlation coefficients (r = 0.974–0.999) and standard error of estimate (SEE) (0.001–0.19). High inter-trial and inter-rater reliability was demonstrated through high intraclass correlation coefficients (ICC = 0.979–1.000), low coefficient of variation (CV = 0.14%), and Bland–Altman plots [28]. However, the latest version of the MySprint app (version 2.0.1) analyses sprint mechanical outputs and FVP's from 30-m sprints, necessitating validation of its reliability and accuracy. While FVP has shown good inter-trial reliability between sprint trials within session (CV = 0.25–6.76%, standard error of measurement = 1.4–4.94%) [9,29], these studies relied on timing photocells to derive split times and failed to declare the specific type of photocells used (i.e., single or dual-beam). This is relevant as single-beam systems are prone to false triggers when determining the exact moment at which the centre of mass passes through the gate, consequently reducing the accuracy and reliability of sprint split times and FVP outputs [30].

Studies evaluating the intra-rater reliability of the current MySprint app for 30-m maximal sprint tests report good-to-excellent agreement (ICC = 0.862–0.984) [25], with low measurement dispersion (CV = 1.3%) indicating strong test-retest reliability [31]. However, variations in methods to address parallax error, such as using vertical marker poles at different distances, likely affect the accuracy of sprint split time and mechanical variable calculations across studies [32]. The app recommends a perpendicular camera distance of 10-m from the midpoint for 30-m sprints, compared to 18-m for the original 40-m sprint analysis [28]. This change in camera placement may impact result consistency and comparability, particularly across studies with unrepresentative samples [31]. A recent study found that video recording devices operating at differing frame rates (i.e., 30 Hz vs. 240 Hz), yielded varying FVP outputs, with greater errors in force-derived parameters compared to velocity, underscoring an important consideration in FVP procedure [33].

Research comparing GPS devices to laser and radar systems for assessing FVP's during linear sprints has yielded mixed results, with a variety of statistical approaches adopted to assess agreement [26,34–37]. An early study comparing GPS (SPI proX, Gpsports, Canberra, Australia; 5 Hz) against a laser system (LDM 301, Jenoptik, Jena, Germany; 100 Hz) found significant inaccuracies in FVP outputs during maximal sprint accelerations (percentage error −5.1 to 2.9%, typical error 5.1–19.2%), recommending against its use for FVP [34]. In contrast, more recent studies indicate moderate-to-good accuracy in the computation of $F_0$, $V_0$, and $FV_{slope}$, alongside intra-system reliability (< 2% bias, ICC = 0.84–0.99) from 10 Hz GPS units (STATSports Apex V3.00 and Catapult Vector S7) when compared to radar for 40 and 50-m sprints [26,35]. These discrepancies likely stem from differences in GPS sampling rates: the earlier study used 5 Hz and 20 Hz units [34], whilst the more recent used 10 Hz [26], exhibiting improved accuracy as sampling rates increased (≥ 10 Hz). Excluding 5 Hz data, both studies support GPS as a valid and reliable tool for FVP during maximal linear sprints [35–37]. Differences in instrument error across profiling techniques likely causes variation in FVP outputs, and the subsequent influence of individual differences on these variations may vary with equipment type [27]. For example, small differences still exist between different GPS device manufacturers and specific models, i.e., STATSports Apex Pro and Catapult Vector S7, which are the most used tracking devices in elite football, meaning outputs from maximal linear sprint testing may not be directly comparable due to inter-unit variation [38].

Simultaneously evaluating the concurrent validity and reliability of FVP methods, such as the MySprint app and GPS units, against the reference radar device during 30-m maximal sprint tests could identify the optimal approach for various contexts, aiding football practitioners in applied settings [19]. Statistical tests examining for agreement

between measurement techniques such as ICC have been critiqued, with recommendations to instead use least products regression analysis to distinguish fixed and proportional bias between methods, alongside Bland-Altman plots [39]. Root mean square error (RMSE) can also be applied to predict the accuracy and subsequent validity of FVP models [40]. Adopting novel research protocols whilst incorporating more suitable statistical analysis techniques can better inform training and rehabilitation programs, potentially reduce injury risk, and enhance sprint acceleration and overall football performance [7,14,18,19,41,42]. FVP generates many sprint mechanical output variables, but the focus of this study will be on $F_0$ (N/kg), $V_0$ (m/s), and $FV_{slope}$, as these are the primary indicators of horizontal force production capacity and encapsulate the overall horizontal force-velocity profile [24], proving highly relevant in both performance and injury contexts [7,18,19]. Leading FVP experts, those behind the conception of the Simple Method [9], have provided a comprehensive overview of key FVP outputs, including relevant definitions and practical interpretations [43].

The aims of the study were to examine the validity and within-session inter-trial, intra- and inter-rater reliability of current sprint force-velocity profiling techniques including the MySprint app, GPS, and radar devices in elite football, and to provide insight into elite footballers' sprint mechanical capabilities.

## Materials and methods

This study comprising of an observational cross-sectional within and between-subjects design followed STROBE reporting guidelines [44] and involved a single testing session consisting of a 30-m maximal sprint testing protocol simultaneously measured using the MySprint app, GPS unit, and radar device. This was conducted at the start of a team training session in the early pre-season on outdoor natural grass, corresponding to 4-days prior to match play [45,46].

### Subjects

Twelve elite youth football players from the academy of an English Premier League team free from illness and injury, volunteered to participate in this study (recruitment started: 26/06/2023, ended: 01/07/2023). Of those volunteers, one player was excluded after failing to complete both sprints, with the remaining eleven players included in further analysis as they had full and standardised data sets for all three equipment types for both sprint trials (male: 11, mean ± SD: age 17.64 ± 1.21 years, height 1.81 ± 0.09 m, body mass 75.10 ± 8.85 kg). Players were classified as "tier 3" – highly trained/national level [47]. Goalkeepers were excluded from the study as they participated in separate training and their in-game movement demands differ to that of outfield players. Players gave written informed consent and, where under 18 years old, written informed consent was sought from the parent or guardian for individuals to participate in this study, which was approved by Lancaster University Medical School (research ethics committee reference: LMS-22–3-Stockdale), and all testing was conducted in accordance with the Declaration of Helsinki.

An *a priori* power analysis was conducted using an online calculator [48], assuming an effect size based on a minimum ICC of 0.70, an expected ICC of 0.95, an alpha level of 0.05, and a power of 80%. The minimum ICC of 0.70 was selected as a conservative threshold for acceptable reliability, while the expected ICC of 0.95 was chosen to reflect the high reliability typically aimed for in profiling measures, in line with prior studies which have reported ICCs within this range [28]. A minimum of 10 participants were required to detect a statistically significant effect when examining for differences in relevant profiling variables [49].

### Testing protocol

Players were instructed to prepare for testing as they would for a regular training session, whilst refraining from excessive exercise, caffeine ingestion, dietary intake, alcohol consumption, and smoking 24-hours prior to testing, and to wear their normal football boots. A familiarisation protocol was not required for the players as they were regularly exposed to maximal sprints, but pilot testing was conducted prior to study commencement to habituate the research team with data

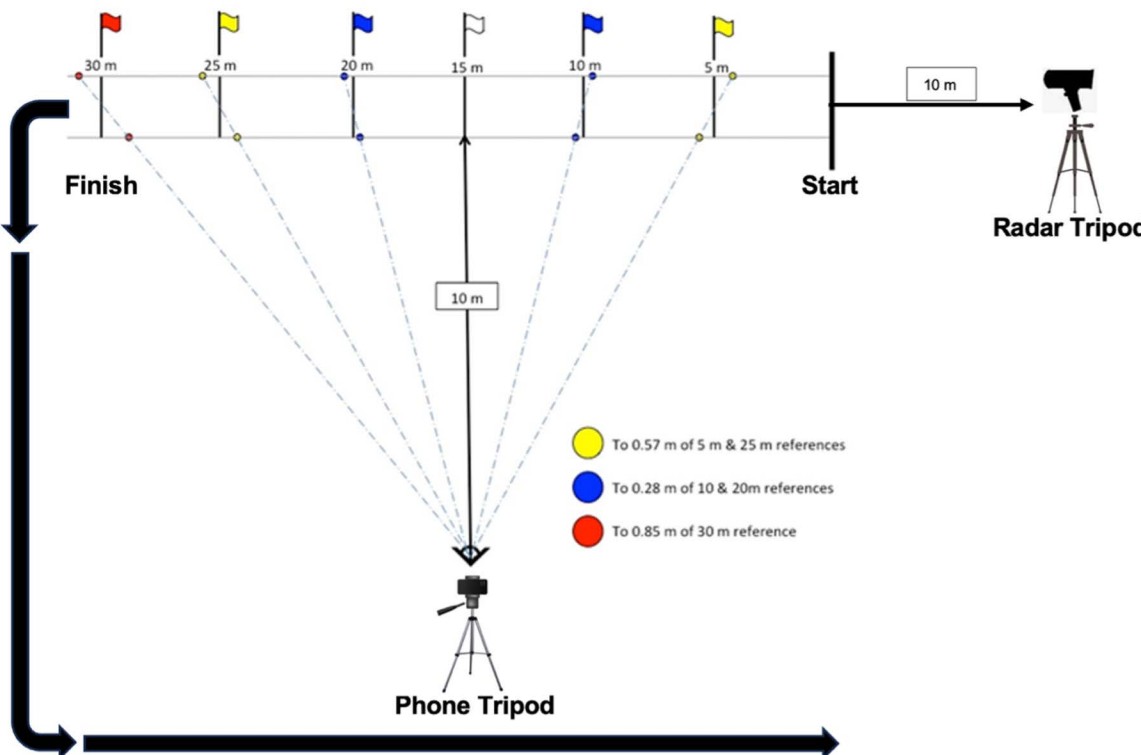

collection techniques. Players completed a standardised 10-minute warmup consisting of a general pulse raiser (jogging), acceleration drills, and two progressive intensity 30–40-m sprints (70 and 80% effort), resembling protocols in previous research [15]. Each player completed two maximal sprints, separated by passive rest (~3-minutes). Sprint starts were standardised, with all players starting from a two-point staggered-stance position [50]. All players were given the instruction to run as fast as they could through the end of the marked track. Maximal sprint testing was recorded for each athlete utilising 10 Hz GPS units (Apex Pro, STATSports, Newry, Ireland). GPS units were placed between the player's scapula using the manufacturers vest, fitted securely to avoid device movement, and were activated 15-minutes prior to the warm-up to ensure good signal quality. The horizontal dilution of precision of the signal and the number of satellites per session were 0.8±0.1 and 17.4±1.2, respectively, which characterised good GPS signal quality [51].

**MySprint app.** Players' sprints were recorded with an iPhone XR (Apple Inc., Cupertino, CA, USA) by the lead academy sports scientist (previous experience in sprint testing video footage capture) using the built-in standard HD video function (1080p at 60 Hz). The linear 30m sprint track was clearly marked using cones and flat markers with six visible vertical marker poles placed along the runway from the start at 5.57m (5m), 10.28m (10m), 15m (15m), 19.72m (20m), 24.43m (25m), and 29.15m (30m) to account for parallax error, as described in similar previous studies [33]. The phone camera was placed on a tripod in landscape orientation, perpendicular to the runway, 10-m away from the 15-m middle vertical marker pole. Tripod height was 1-m to align approximately level with the players' hip centre of mass [28]. Players' shorts contrasted the colour of vertical marker poles, making it easier to locate the hip centre of mass against them. Fig 1 details the equipment setup for the 30-m maximal sprint testing protocol.

**Fig 1. 30-m maximal sprint testing protocol equipment setup.** Arrows at finish denote direction players were instructed to return to the start – exiting the radar field and passing behind the camera to avoid confounding.

Sprint videos were subsequently imported into and analysed using the MySprint app to directly determine sprint mechanical output variables and subsequent FVP's (version 2.0.1) [45]. The start of the sprint was defined as the moment at which body movement started, specifically the first instance of lower-limb motion from standstill preceding the onset of the sprint start (detected via visual inspection). The hip centre of mass was located at the centre of the pelvis, identifiable during video footage analysis. The frames were then selected in which each players' hip centre of mass was aligned with each of the six vertical marker poles. Misidentification of the start, and frame in which the players' hip centre of mass is aligned with (crosses) each of the six vertical marker poles, has been reported to induce error and alter the resultant sprint mechanical output variables [25]. Player's body mass, stature, and split times were used by the MySprint app to calculate $F_0$ (N/kg), $V_0$ (m/s), and $FV_{slope}$, ultimately determining individual FVP's, using previously validated formulas [9]. Individual differences in body mass were accounted for by incorporating body mass into metric calculations, i.e., N/kg.

Intra-rater reliability was assessed by analysing the fastest sprint for each player using the MySprint app. The resulting FVP data were recorded, and the analysis process in the app was repeated (analyse, delete, and re-analyse) to evaluate variation in the results when the same rater performed the analysis multiple times. These analysis repeats were conducted immediately following the testing session. Regarding inter-rater reliability, the primary rater (lead author) was highly experienced in sprint testing video footage capture and the use of the MySprint app whereas the second rater (academy sports scientist) had previous experience in sprint testing video capture but not specifically using the MySprint app (i.e., novice). Before the study, both raters underwent basic app orientation and practice analysis. This included a detailed 30-minute example session led by the primary rater, who guided the second rater through the analysis process. The session followed the same procedures and instructions outlined in the previous section.

**Radar device.** The radar device (Stalker ATS Pro II; Applied Concepts, Plano, TX, USA) measured instantaneous velocity at a sampling rate of 46.875 Hz and was placed on a tripod 10-m behind the athletes at a height of 1-m, corresponding approximately to the players' hip centre of mass [46]. Recording using this device was enacted by the lead author who was experienced in radar device data capture. During the session, data were recorded using a laptop running Stalker ATS System™ software (Version 5.1.1, Applied Concepts, Inc., Texas, USA). Radar data acquisition started once the player was in the start position, prior to the moment at which body movement started to fully capture the sprint start and ended once the player had passed the finish. The raw data capture file for each maximal sprint was then manually saved to the computer. Following the testing session these files were manually processed in the software system by deleting all data recorded prior to the start and after the finish of each sprint and classifying all trials as 'acceleration runs' thereby forcing the start of the velocity-time curve through the zero point [52]. Filtering type selected was 'Dig Medium', concomitant with research investigating maximal sprint performance in football [53]. The original files (.rda) were converted into a different format (.rad or.xlsx), the start of the sprint was identified as significant increase on the speed plot, i.e., >0.2 m/s (0.72 km/h), to max velocity reached. Excel scatter plots (smooth lines and markers) were created for each sprint to visualise both maximal sprints performed by each player. Time (s) and instantaneous speed (km/h) data were copied and pasted into the radar force-velocity spreadsheet to calculate FVP outputs [54].

**GPS unit.** Individual GPS units (Apex Pro, STATSports, Newry, Ireland; 10 Hz) were continuously recording during the 30-m maximal sprint testing protocols. Following the session, data were downloaded using the manufacturers software (Sonra, STATSports, Newry, Ireland), isolated maximal sprint efforts were 'clipped', and custom Microsoft Excel export created containing relevant metrics required for FVP calculation (time, speed, acceleration), for both maximal sprints performed during each testing protocol. It is important to note that the raw GPS data may have had a level of filtering applied by the manufacturer, but unfortunately the nature or degree of any such treatment is not disclosed to the end-user [38]. Time data were converted from 24HR format to seconds at 0.1s intervals. The start of the sprint was identified as significant increase on the speed plot, i.e., >0.2 m/s, to max velocity reached. Excel scatter plots with smooth lines and markers were created for each sprint to visualise both maximal sprints performed by each player. Time (s) and instantaneous speed (m/s) data were copied and pasted into the GPS force-velocity spreadsheet to calculate FVP outputs [14,54].

Standardised >0.2 m/s increase in speed threshold was used to denote the sprint start for both the radar and GPS units. For most players the root-mean-square error value corresponding to these cut-offs was <0.2 thus indicating good reliability and

validity [55]. Most studies in this area have also incorporated these thresholds, enabling valid and reliable comparisons in relevant sprint mechanical output variables [28,52]. The higher sampling rate of the radar device prevented significant differences in speed between samples meaning a higher speed threshold, i.e., >0.5–1 m/s would have made it difficult to determine the actual sprint start. Acceleration ($m/s^2$) thresholds were not used as these devices can be unreliable at measuring acceleration at low speed, and these secondary values are derived from the primary raw speed data anyway [56,57]. GPS and radar sprint mechanical output variable calculation spreadsheets can be accessed in the supporting information section (S1 and S2 Files).

## Statistics

Statistical analyses were performed using Jamovi (version 2.3.28). Shapiro-Wilk tests determined data did not follow a normal distribution for all metrics between equipment types, necessitating the use of non-parametric tests. Intra- and inter-rater reliability using the MySprint app were determined using the Wilcoxon Matched Pairs test, RMSE, and CV, alongside ICC (3,1) for intra-rater reliability and ICC (3,2) for inter-rater reliability, with 95% confidence intervals (CI) [58,59]. Differences in $F_0$ and $V_0$ between equipment types were examined using a Kruskal-Wallis test, and level of agreement between devices using Bland-Altman plots (±1.96 SD). Inter-trial reliability was analysed using the Wilcoxon Matched Pairs test, ICC (3,1), RMSE, and CV. Statistical significance was $p < 0.05$. Dwass-Steel-Critchlow-Fligner pairwise comparison was applied to examine for differences between two specific equipment types if the Kruskal-Wallis test reported significance, and epsilon squared ($\varepsilon^2$) effect sizes were calculated; 0.01–0.059 (small effect), 0.06–0.139 (moderate effect) and ≥ 0.14 (large effect) [60]. For the Wilcoxon Matched Pairs tests, rank biserial correlation ($r_{rb}$) effect sizes were calculated and interpreted as small (0.01–0.19), medium (0.20–0.49), and large (> 0.50) [61]. ICC (reliability) was interpreted as: <0.5=poor; 0.5–0.75=moderate; 0.75–0.9=good; and > 0.90=excellent [62]. CV was calculated by dividing the standard deviation by the mean and multiplying by 100 and was interpreted as good when < 10% [63]. RMSE was interpreted as good when < 0.2 [55]. Descriptive data are presented as raw values, means ± SD, medians, or confidence intervals where appropriate.

## Results

### Descriptive data

Table 1 outlines the FVP outputs from 30-m maximal sprint testing ($F_0$, $V_0$, and $FV_{slope}$), for different equipment types (MySprint app, GPS, and radar device).

### Concurrent validity of the MySprint app and GPS

There were significant differences in $F_0$, $V_0$, and $FV_{slope}$ between the MySprint app, GPS units, and radar device ($p < 0.05$), with large effect sizes reported ($\varepsilon^2 = 0.621$, 0.684, and 0.781, respectively). Pairwise comparisons reported significant differences with the MySprint app demonstrating elevated $F_0$ in conjunction with lower $V_0$, and more negative $FV_{slope}$ compared to the radar device ($p < 0.05$), with large effect sizes reported ($\varepsilon^2 = 0.683$, 0.513, and 0.482). There were no significant differences in $F_0$ or $V_0$ ($p > 0.05$), with large effect sizes reported ($\varepsilon^2 = 0.271$ and 0.281), between the GPS units and radar device. Bland-Altman limits of agreement (±1.96 SD) between GPS and radar were ±0.194 N/kg for $F_0$ and ± 0.09 m/s for $V_0$, with data points close to 0 and consistent variability, indicating no proportional bias. $F_0$ and $V_0$ for each participant deriving from the different equipment are shown in Fig 2. Differences in $FV_{slope}$ between equipment is demonstrated by the profiles superimposed in Fig 3.

Table 1. FVP outputs from 30-m maximal sprint testing. Data presented as means ± SD.

| | $F_0$ (N/Kg) | $V_0$ (m/s) | $FV_{slope}$ |
|---|---|---|---|
| **MySprint app** | 10.10 ± 0.88 | 7.82 ± 0.30 | −0.119 ± 0.017 |
| **GPS** | 7.39 ± 0.49 | 8.80 ± 0.33 | −0.062 ± 0.004 |
| **Radar** | 7.58 ± 1.10 | 8.89 ± 0.30 | −0.081 ± 0.013 |

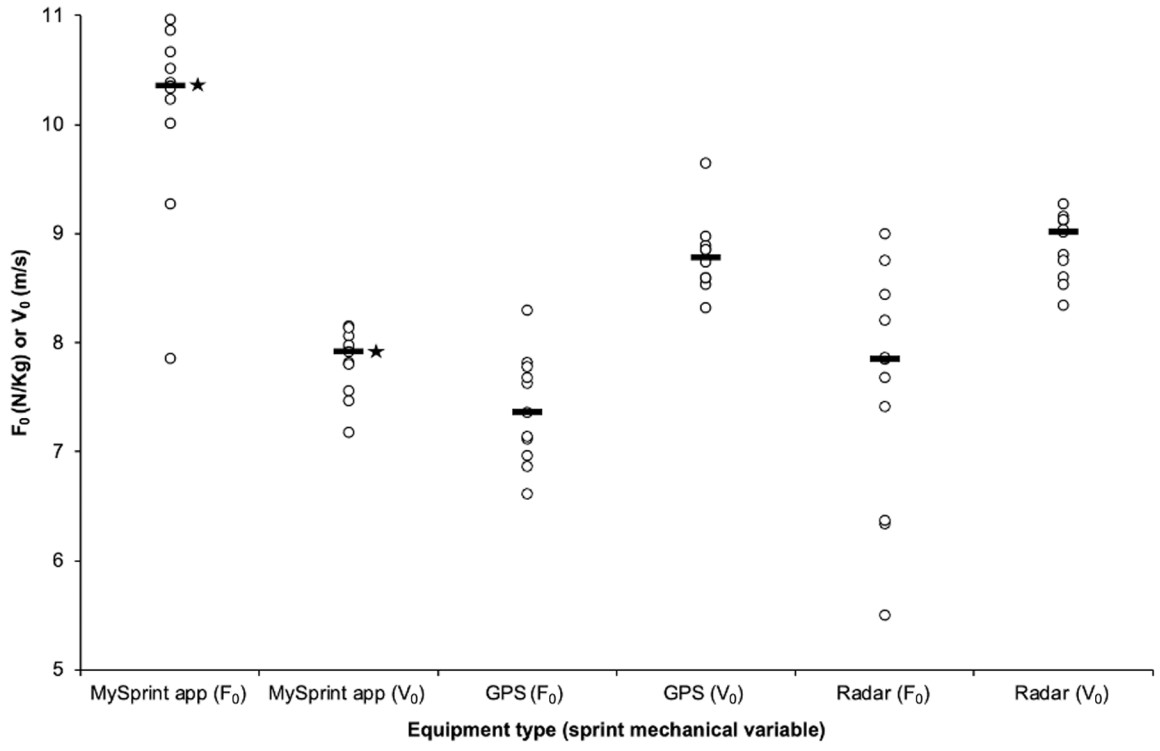

**Fig 2. $F_0$ and $V_0$ between equipment types.** Raw data presented for each participant with * indicating a significant difference in the median compared to the radar device and GPS, and black line markers denoting the median for each group.

### Inter-trial reliability

There were no significant differences in $F_0$ or $V_0$ ($p > 0.05$) between the two sprint trials performed by each player for all equipment types used, with the following effect sizes ($r_{rb}$): MySprint (0.047, 0.523), GPS (0.236, −0.163), and radar (0.785, −0.777). ICC reported poor-to-good levels of agreement for the MySprint app ($F_0$ ICC = 0.285 and $V_0$ ICC = 0.865), poor-to-moderate levels of agreement for the radar device ($F_0$ ICC = 0.324 and $V_0$ ICC = 0.548), and poor levels of agreement for the GPS ($F_0$ ICC = 0.064 and $V_0$ ICC = 0.437) between the two sprint repeats. RMSE and CV values were as follows: MySprint ($F_0$ and $V_0$ were 0.72 N/Kg and 3.76%, 0.21 m/s and 1.48%), radar ($F_0$ and $V_0$ were 1.11 N/Kg and 8.66%, 0.30 m/s and 2.22%), GPS ($F_0$ and $V_0$ were 1.20 N/Kg and 9.59%, 0.40 m/s and 2.64%). Variation in $F_0$ and $V_0$ between sprint repeats for each player, using each equipment type are shown in Fig 4.

### Intra-rater reliability

There was no significant difference in $F_0$ ($W = 25$) or $V_0$ ($W = 12$) ($p > 0.05$) when the MySprint analysis procedure was repeated by the same rater, with medium-to-large effect sizes reported ($r_{rb} = -0.242$ and $-0.636$). RMSE and CV values for $F_0$ and $V_0$ were 0.19 N/Kg and 2.9%, 0.07 m/s and 0.54%. ICC demonstrated good and excellent levels of agreement in the values of $F_0$ and $V_0$ between repeated MySprint analysis (ICC = 0.832, CI = 0.508–0.951, and ICC = 0.976, CI = 0.871–0.994, respectively). Consistency of the MySprint app analysis procedure in measuring $F_0$ and $V_0$ is visualised using a univariate scatter plot for paired data in Fig 5.

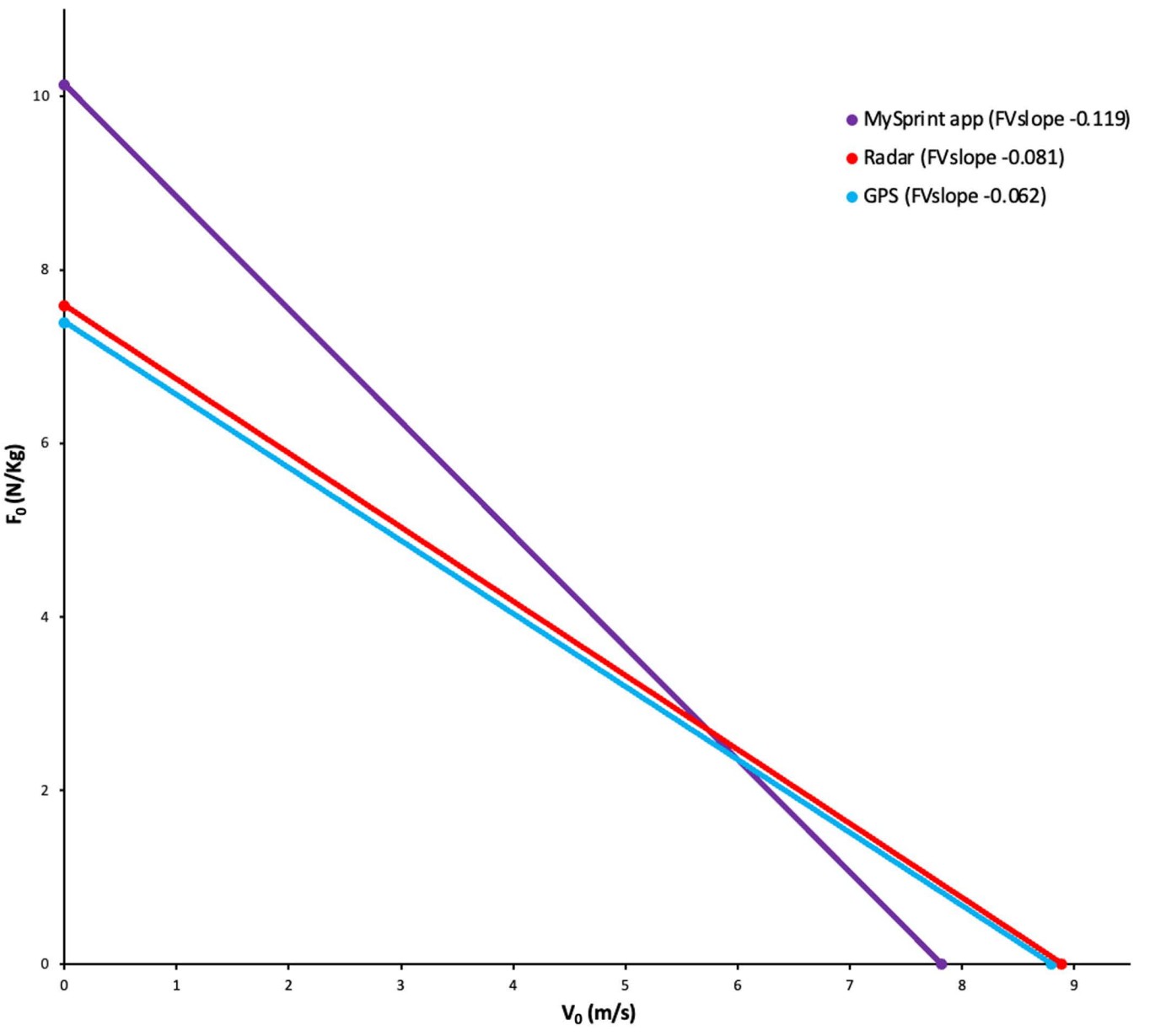

**Fig 3. Force-velocity profiles (FV$_{slope}$) between equipment types.**

### Inter-rater reliability

There was no significant difference in $F_0$ (W = 26) or $V_0$ (W = 12.5) ($p > 0.05$) between the two raters performing the MySprint app analysis, with medium-to-large effect sizes reported ($r_{rb}$ = −0.212 and −0.621). RMSE and CV values for $F_0$ and $V_0$ were 0.18 N/Kg and 1.04%, 0.06 m/s and 0.42%. ICC demonstrated excellent levels of agreement in the values of $F_0$ and $V_0$ between the two raters (ICC = 0.986, CI = 0.950–0.996, and ICC = 0.988, CI = 0.957–0.997). Bland-Altman analysis showed limits of agreement (±1.96 SD) of ± 0.03 N/kg for $F_0$ and ± 0.008 m/s for $V_0$, with most data points near 0 and consistent variability across the force-velocity spectrum, highlighting an absence of proportional bias. The level of agreement in $F_0$ (**a**) and $V_0$ (**b**) between the two raters is visualised using Bland-Altman plots in Fig 6.

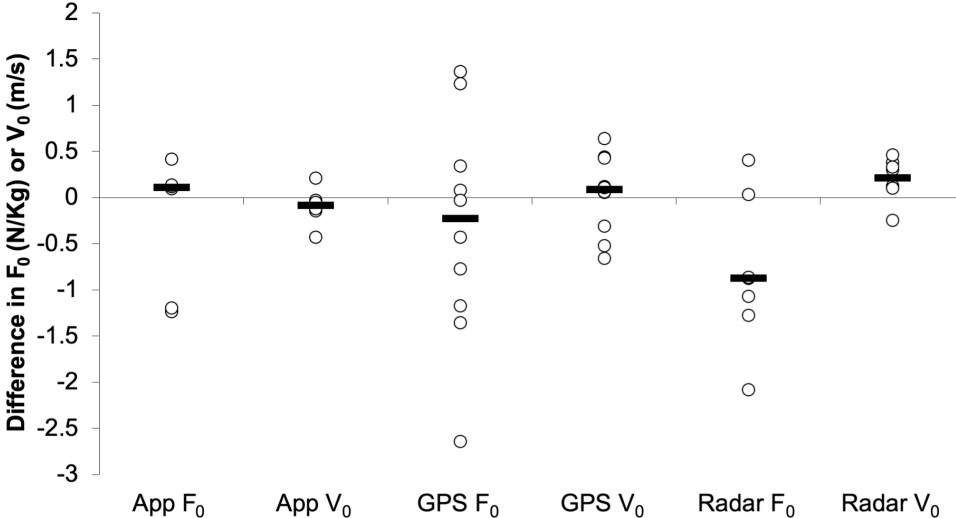

**Fig 4. Difference in $F_0$ and $V_0$ between sprint repeats for each participant, derived from each equipment type, with black line markers denoting each group's median difference.**

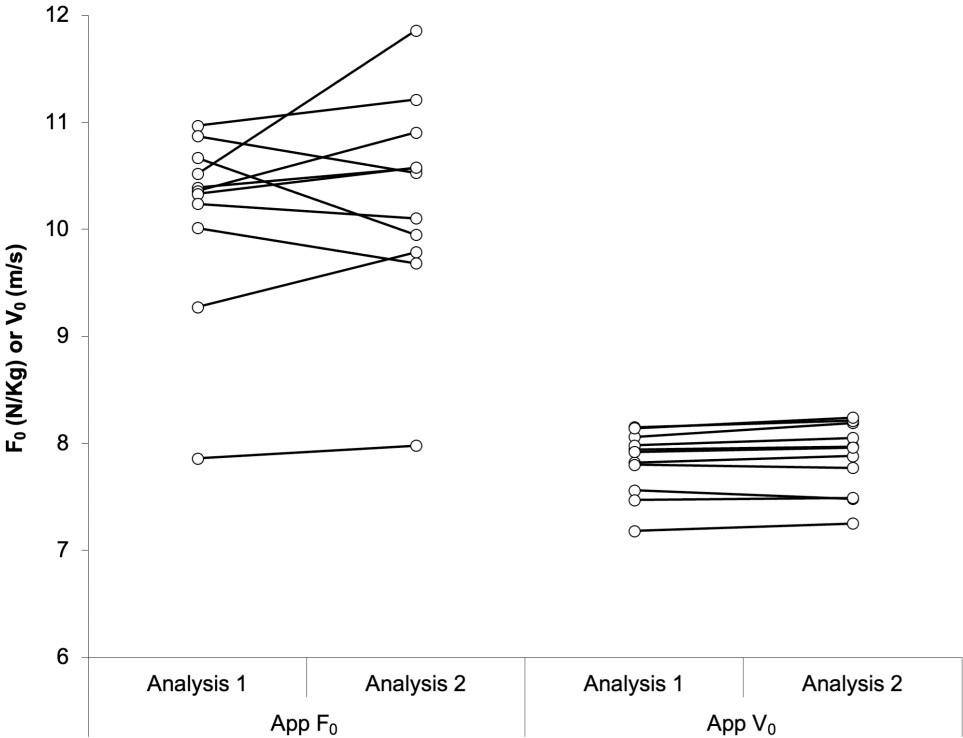

**Fig 5. Descriptive univariate scatter plot for paired data demonstrating consistency in the measurement of $F_0$ and $V_0$ between intra-rater analysis repeats.**

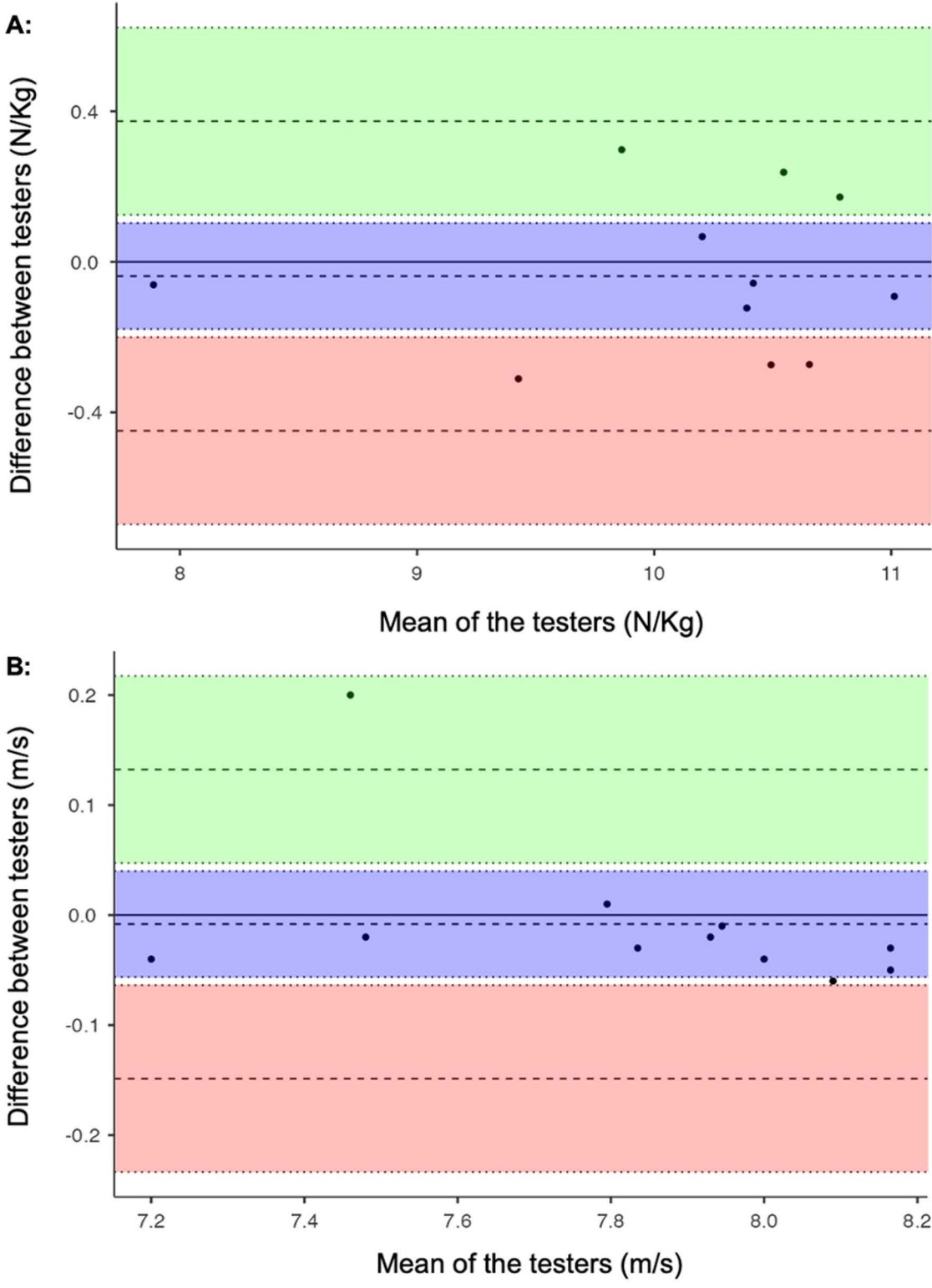

**Fig 6. Bland-Altman plots for $F_0$ (A) and $V_0$ (B) between raters.** The central dotted line represents the absolute average difference (bias) between raters, whilst the upper and lower dotted lines represent limits of agreement ($\pm$1.96 SD).

## Discussion

The aim of the study was to examine the validity and reliability of current force-velocity profiling techniques in elite football, and to provide insight into elite footballers' sprint mechanical capabilities. The study reported high intra- and inter-rater reliability when using the MySprint app to derive FVP's from 30-m maximal sprint testing, but this coincided with reduced validity purported by higher bias alongside inflated $F_0$ and reduced $V_0$ compared to the GPS units and radar device. Overall, GPS is the most valid and reliable force-velocity profiling technique, which most closely aligns with the reference radar device. As such, these findings suggest that MySprint and GPS profiling techniques are valid and reliable, as the former demonstrated good reliability but poor validity, whereas the latter exhibited acceptable validity and reliability whilst most closely aligning with the reference radar device.

### Descriptive data

Regarding the FVP outputs from 30-m maximal sprint testing (Table 1), the MySprint app $F_0$ is higher, but $V_0$ lower, GPS $F_0$ is comparable although $V_0$ is slightly higher [14], whereas the radar device derived $F_0$ and $V_0$ [10,12] align with normative data for FVP outputs deriving from previous maximal sprint testing conducted using elite male footballers.

### Concurrent validity of the MySprint app and GPS

GPS demonstrated superior validity compared to the MySprint app, against the reference radar device. Significant differences in $F_0$ and $V_0$ were observed between the MySprint app, GPS, and radar devices, with large effect sizes reported. Pairwise comparisons revealed that the MySprint app produced higher $F_0$ in conjunction with lower $V_0$ compared to the radar device ($p < 0.05$, Fig 2), and $FV_{slope}$ varied across the MySprint app, GPS, and radar device, with the former demonstrating the most negative slope (Fig 3). This aligns with prior research identifying low agreement in sprint split times and derived mechanical variables between the MySprint app and other instruments [25,31]. Conversely, earlier studies reported strong agreement between the MySprint app and radar [28]. The differences observed in this study are likely due to the MySprint app's sensitivity to minor inaccuracies in determining sprint start time. Such errors could lead to inflated $F_0$ and reduced $V_0$ if the visual identification of movement initiation is incorrect [25]. However, no significant differences in $F_0$ or $V_0$ ($p > 0.05$) were found between GPS and radar, supporting earlier findings of moderate-to-perfect correlations between these devices, with higher GPS sampling rates ($\geq 10$ Hz) reducing error bias in force-velocity variables [34,35]. Bland-Altman plots examining for differences in FVP outputs between the GPS and radar devices demonstrated data points close to zero and consistent variability, indicating no proportional bias. Caution is warranted when comparing results across studies, as GPS validity against radar devices varies between manufacturers and models, i.e., GPS units from STATSports demonstrated moderate-to-good validity, whereas Catapult devices showed good validity [26]. GPS is the most valid against the reference radar device for 30-m sprint testing, but practitioners must carefully consider the specific GPS model and manufacturer, as validity can differ between devices.

### Inter-trial reliability

GPS and radar achieved comparable inter-trial reliability. There were no significant differences in $F_0$ or $V_0$ ($p > 0.05$) across the two sprints performed by each player using any of the equipment types tested (MySprint app, GPS, and radar), indicating that all systems demonstrated reliability in measuring FVP outputs across repeated trials (Fig 4). Evaluating $F_0$ and $V_0$ between the two sprints, ICC revealed poor-to-good agreement for the MySprint app, poor agreement for GPS, and poor-to-moderate agreement for the radar device. RMSE and CV values for the MySprint app's $F_0$ and $V_0$ measurements were interpreted as "poor" (RMSE $> 0.2$) and "good" (CV $< 10\%$), respectively. Similarly, GPS-derived outputs for $F_0$ and $V_0$ were rated "poor" for RMSE but "good" for CV. Radar device outputs followed the same trend, with "poor" RMSE but "good" CV values. These findings align with earlier research on the MySprint app, which reported minimal variation in FVP

outputs across trials (CV = 0.14%) [28]. Comparatively, radar-based studies reported greater variability and slightly lower agreement in mechanical outputs between sprints (CV = 1.4–11%; ICC = 0.75–0.99), though these levels were still considered indicative of good reliability [52,64]. Similarly, studies on GPS demonstrated comparable reliability when assessing FVP outputs between repeats (CV = 0.1–11.53%) [36,65]. These findings reinforce that radar and GPS devices achieve comparable reliability in measuring FVP outputs across repeated sprints.

A noteworthy trend in the current study is the higher reliability and agreement levels in $V_0$ compared to $F_0$ across all devices between trials. This pattern is consistent with the aforementioned FVP studies, which have reported greater reliability for FVP outputs towards the velocity side of the spectrum, as opposed to lower reliability for force-dependent variables. Variability may stem from equipment proving more inconsistent in the measurement of high levels of force generated at the start of sprint accelerations, compared to the relatively stable measurement towards maximal velocity [14]. From a practical perspective, the GPS unit demonstrated variability of 0.69 N/kg for $F_0$ and 0.23 m/s for $V_0$ between sprint repeats, well below the threshold of a 1 N/kg decrease in $F_0$ associated with a 2.67-fold increased risk of hamstring injury [18]. These results warrant caution as the combined effect of several levels of error, i.e., biological, intra-rater, and equipment, means it is difficult to definitively attribute divergence in FVP outputs between trials.

### Intra-rater reliability

The MySprint app demonstrated intra-rater reliability which was stronger in the computation of $V_0$ compared to $F_0$. No significant differences in $F_0$ or $V_0$ were observed when the MySprint analysis was repeated by the same rater using video footage from 30-m maximal sprint testing. RMSE and CV values for both variables were classified as "good" (RMSE < 0.2, CV < 10%), demonstrating high accuracy and low measurement dispersion. ICC values also indicated good-to-excellent agreement for $F_0$ and $V_0$, confirming that a single rater can reliably use the MySprint app to calculate FVP outputs from 30-m sprint protocols. These findings align with previous studies on the app's intra-rater reliability for sprint split time and mechanical variable computation, reporting similar agreement levels for experienced (ICC = 0.984) and non-experienced (ICC = 0.862) raters, alongside low measurement dispersion (CV = 1.307%) and high test-retest reliability [25,31]. The lower intra-rater agreement for $F_0$ (ICC = 0.832) compared to $V_0$ (ICC = 0.976) (Fig 5) may result from the subjective nature of manual frame selection in the MySprint app. Misidentification of the onset of body movement during visual inspection could alter sprint split time calculations, leading to variability in resultant $F_0$ [25].

### Inter-rater reliability

The MySprint app displayed strong inter-rater reliability. There were no significant differences in $F_0$ or $V_0$ between two raters conducting the MySprint app analysis for 30-m maximal sprint testing. Both RMSE and CV values were classified as "good" (RMSE < 0.2, CV < 10%), indicating high precision and minimal variability in measurements. ICC values demonstrated excellent agreement, confirming robust inter-rater reliability. This ensures that different raters can confidently use the MySprint app interchangeably to calculate FVP outputs during 30-m sprint protocols. Bland-Altman plots examining for variation between the different raters demonstrated data points close to zero and consistent variability, indicating no proportional bias (Fig 6). Prior research on inter-rater reliability for split time measurement using the MySprint app in 30 and 40-m sprints supports these findings (ICC = 0.969–1.000) [25,28,31]. The present study's ICC values (0.986–0.988) are similarly high, although rater experience was unclear in two of the comparison studies, which could lead to discrepancies in sprint split time and mechanical variable calculations between raters with varying levels of proficiency. The app's strong inter-rater reliability facilitates use by multiple practitioners, enhancing its practicality and accessibility in elite football environments where efficient and consistent data collection is critical. However, this reliability coincided with reduced validity purported by higher bias alongside inflated $F_0$ and reduced $V_0$ compared to both the GPS units and radar device.

## Limitations

GPS units can potentially prove unreliable due to noise in data, particularly at high speeds [56], and may lack the necessary accuracy and sensitivity for capturing accelerations at low speeds (< 2 m/s) and across different units [57]. It is recommended that individuals use the same GPS device during testing to reduce inter-unit variability. Force-velocity profiles derived from GPS data depend on linear fitting, influenced by the quality of the signal which is affected by several factors [51]. Higher sampling rates (≥ 10 Hz), such as those employed by the GPS in the current study, have been reported to significantly enhance relevant measurement reliability. Potential false indications of the MySprint app's validity/reliability (or lack of) due to the subjective nature of manual frame selection – evidenced by over-inflation of $F_0$ and subsequent reduction in $V_0$ sprint mechanical output values, perhaps suggests that the moment at which body movement started (sprint start) was consistently incorrectly identified. Future studies should aim to include larger more representative samples of sufficient size to provide adequate power necessary to derive more reliable and statistically sound conclusions, further acting to increase the generalisability of findings to the elite football environment.

## Conclusion

GPS was valid and demonstrated strong agreement against the reference radar device when determining FVP outputs from 30-m maximal sprint testing. The MySprint app demonstrated good intra- and excellent inter-rater reliability, but this coincided with reduced validity purported by higher bias alongside inflated $F_0$ and reduced $V_0$ compared to the GPS units and radar device. All equipment (MySprint app, GPS, and radar device) indicated similar levels of moderate inter-trial reliability in the measurement of FVP outputs between sprint repeats, with GPS displaying values most closely representing the radar device. Given the time-consuming and possibly error-inducing nature of manual 2D analysis and frame identification, it is recommended to use the faster GPS or radar methods instead. GPS is the most valid and reliable technique when assessing FVP outputs and force-velocity profiles from 30-m maximal sprints, demonstrating 0.69 N/Kg and 0.23 m/s inter-trial variability in $F_0$ and $V_0$ respectively, falling adequately below the 1 N/Kg decrease of $F_0$ associated with 2.67 times higher risk of sustaining a new hamstring injury [18]. Therefore, GPS can be considered an alternative and potentially more accessible method to provide FVP outputs in elite football.

## Supporting information

**S1 File. GPS FVP calculation spreadsheet.**
(XLSX)

**S2 File. Radar FVP calculation spreadsheet.**
(XLSX)

**S3 File. Spreadsheet containing raw data.**
(XLSX)

## Author contributions

**Conceptualization:** Robert Alexander Stockdale.

**Data curation:** Robert Alexander Stockdale.

**Formal analysis:** Robert Alexander Stockdale.

**Funding acquisition:** Robert Alexander Stockdale.

**Investigation:** Robert Alexander Stockdale.

**Methodology:** Robert Alexander Stockdale, Thomas Dos'Santos, Kevin McDaid, Christopher J. Gaffney, Timothy J. Barry.

**Project administration:** Robert Alexander Stockdale.

**Resources:** Robert Alexander Stockdale.

**Software:** Robert Alexander Stockdale.

**Supervision:** Christopher J. Gaffney, Timothy J. Barry.

**Validation:** Robert Alexander Stockdale.

**Visualization:** Robert Alexander Stockdale.

**Writing – original draft:** Robert Alexander Stockdale, Christopher J. Gaffney, Timothy J. Barry.

**Writing – review & editing:** Robert Alexander Stockdale, Thomas Dos'Santos, Kevin McDaid, Philip Nagy, Christopher J. Gaffney, Timothy J. Barry.

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
