## [Decision Letter · Decision Letter 0]

31 Mar 2025

Dear Dr. Stockdale,

Thank you for submitting your manuscript to PLOS ONE. After careful consideration, we feel that it has merit but does not fully meet PLOS ONE’s publication criteria as it currently stands. Therefore, we invite you to submit a revised version of the manuscript that addresses the points raised during the review process.

We look forward to receiving your revised manuscript.

Kind regards,

Julio Alejandro Henriques Castro da Costa

Academic Editor

PLOS ONE

“RAS was a part-time paid employee at Burnley F.C. at the time of data collection. This ceased after June 2023, prior to the writing of the manuscript.”

Reviewers' comments:

Reviewer's Responses to Questions

**Comments to the Author**

1. Is the manuscript technically sound, and do the data support the conclusions?

Reviewer #1: Yes

Reviewer #2: Yes

2. Has the statistical analysis been performed appropriately and rigorously?

Reviewer #1: Yes

Reviewer #2: Yes

3. Have the authors made all data underlying the findings in their manuscript fully available?

Reviewer #1: Yes

Reviewer #2: Yes

4. Is the manuscript presented in an intelligible fashion and written in standard English?

Reviewer #1: Yes

Reviewer #2: Yes

Reviewer #1: It is an interesting, well-conducted study related to sports performance. The discussions contain adequate integration of this study in the context of medical literature. Iconography is relevant. I particularly appreciate the fairness on the section related to study’s limitations.

Reviewer #2: The introduction is well written and covers all important aspects that are needed in order to understand why this research is needed.

Methods:

- L177-180: Even though a power calculation was performed, it is not clear how the effect sizes used for calculation were determined. Was a similar study used as a reference? If yes, please cite this study. If not, please describe why you chose these values.

Results and the discussion are clear and concise. They also cover all relevant aspects and are written well.

**Do you want your identity to be public for this peer review?** For information about this choice, including consent withdrawal, please see our Privacy Policy

Reviewer #1: **Yes: ** Diana Ciubotariu

Reviewer #2: No

---

## [Author Response · Author response to Decision Letter 1]

26 Apr 2025

Point-by-point response to the additional requirements outlined in the decision email:

1. Manuscript formatting: The manuscript has been revised to meet PLOS ONE's formatting and style requirements.

2. Consent from minors: The manuscript already includes a statement regarding informed consent from participants and their guardians:

“Players gave written informed consent and, where under 18 years old, written informed consent was sought from the parent or guardian for individuals to participate in this study, which was approved by Lancaster University Medical School (research ethics committee reference: LMS-22-3-Stockdale), and all testing was conducted in accordance with the Declaration of Helsinki.”

3. Financial disclosure: This section has been clarified as follows:

a. Robert Alexander Stockdale (lead/corresponding author) was a part-time paid employee at Burnley F.C. at the time of data collection (First Team Research & Development Assistant) who received a monthly salary from the club; but this employment ended in June 2023, prior to manuscript preparation.

b. No financial or material funding was received specifically for this study.

c. The funders had no role in the study design, data interpretation, manuscript preparation, or decision to publish.

d. The lead academy sports scientist at Burnley F.C. contributed to data collection and analysis using the MySprint app.

e. To confirm, the authors received no specific funding for this work.

In summary, the lead author worked for Burnley F.C. during data collection but not during writing; there was no direct funding/influence from the club; and while a club staff member assisted data collection, research was independently conducted.

4. Abstract: The abstract in the manuscript has been revised to match the version provided in the online submission form.

5. References: The reference list has been reviewed and updated as follows:

a. Reference 24: Reformatted per PLOS ONE guidelines.

b. Reference 26: Reformatted per PLOS ONE guidelines.

c. Reference 50: Reformatted as an app citation in accordance with guidelines.

d. Reference 54: Reformatted per PLOS ONE guidelines.

e. Reference 61: Reformatted appropriately.

6. Reviewer #2 – Methods (Lines 177–180):

In response to the reviewer’s request for clarification regarding the effect sizes used in the power analysis, we have added an explanation to the manuscript. Specifically, we now justify the choice of ICC values with reference to a relevant profiling study and conventional thresholds for acceptable and excellent reliability.

---

## [Decision Letter · Decision Letter 1]

16 May 2025

Validity and Reliability of Sprint Force-Velocity Profiling in Elite Football: Comparison of MySprint, GPS, and radar devices.

PONE-D-25-13363R1

Dear Dr. Stockdale,

We’re pleased to inform you that your manuscript has been judged scientifically suitable for publication and will be formally accepted for publication once it meets all outstanding technical requirements.

Kind regards,

Julio Alejandro Henriques Castro da Costa

Academic Editor

PLOS ONE

Additional Editor Comments (optional):

Reviewers' comments:

Reviewer's Responses to Questions

**Comments to the Author**

Reviewer #1: All comments have been addressed

2. Is the manuscript technically sound, and do the data support the conclusions?

Reviewer #1: Yes

3. Has the statistical analysis been performed appropriately and rigorously?

Reviewer #1: Yes

4. Have the authors made all data underlying the findings in their manuscript fully available?

Reviewer #1: (No Response)

5. Is the manuscript presented in an intelligible fashion and written in standard English?

Reviewer #1: Yes

Reviewer #1: I consider that the manuscript, in the current form, is suitable for publication. It respects the requirement of the journal and it properly answered reviewers request.

**Do you want your identity to be public for this peer review?** For information about this choice, including consent withdrawal, please see our Privacy Policy

Reviewer #1: **Yes: ** Diana Ciubotariu

---

## [Editor Report · Acceptance letter]

PONE-D-25-13363R1

PLOS ONE

Dear Dr. Stockdale,

I'm pleased to inform you that your manuscript has been deemed suitable for publication in PLOS ONE. Congratulations! Your manuscript is now being handed over to our production team.

Kind regards,

on behalf of

Dr. Julio Alejandro Henriques Castro da Costa

Academic Editor

PLOS ONE